# The Lymphatic Vascular System in Extracellular Vesicle-Mediated Tumor Progression

**DOI:** 10.3390/cancers16234039

**Published:** 2024-12-02

**Authors:** Pragati Lodha, Alperen Acari, Jochen Rieck, Sarah Hofmann, Lothar C. Dieterich

**Affiliations:** 1European Center for Angioscience (ECAS), Medical Faculty Mannheim, Heidelberg University, 68167 Mannheim, Germany; 2Mannheim Institute for Innate Immunoscience (MI3), Medical Faculty Mannheim, Heidelberg University, 68167 Mannheim, Germany; 3Heidelberg Bioscience International Graduate School (HBIGS), Faculty of Bioscience, Heidelberg University, 69120 Heidelberg, Germany

**Keywords:** lymph node, pre-metastatic niche, tumor immunity, lymphatic drainage, lymphatic transport

## Abstract

Cancer growth relies on interactions between cancerous and normal cells. Extracellular vesicles, nanometer-sized particles released by cells, play a big role in these interactions by carrying signals that can help the cancer spread and escape the body’s defenses. These vesicles travel from the tumor to other parts of the body, sometimes preparing distant organs to receive cancer cells. The lymphatic system, a network of vessels different from blood vessels that primarily serves as a drainage system, is especially important for transporting these vesicles to nearby lymph nodes and beyond. Recent research shows that tumor-draining lymph nodes are key spots where extracellular vesicles can influence other cells to support cancer progression. Understanding exactly how extracellular vesicles move through the lymphatic system and alter the microenvironment of the lymph node could open up new ways to target and treat cancer.

## 1. Overview of the Lymphatic Vascular Network

The lymphatic vascular network represents a one-way conduit, whereas blood vessels form a circulatory system. A significant part of the liquid blood fraction, the plasma, continuously leaks from the arterial side of the capillary beds into the interstitial space. Thus, constant drainage of excess fluid and macromolecules by initial lymphatic vessels is required to maintain fluid homeostasis of the body [1,2,3]. Blind-ended initial lymphatic vessels form a dense network in the interstitial connective tissue of almost all organs and tissues (Figure 1A,B). Lymphatic endothelial cells (LECs) that delineate the lumen of initial lymphatics are interconnected by button-shaped adhesive junctions. Junctional foci are positioned several µm apart from each other, resulting in valve-like flaps through which interstitial fluid seeps from the interstitium into the vessel lumen [4]. The initial lymphatic vessels converge to intermediate segments (precollectors) and finally to collecting vessels (collectors), which transport the lymph to regional lymph nodes (LNs, Figure 1A).

The composition of the lymph initially resembles that of the interstitial fluid, containing plasma proteins, tissue-derived solutes and molecules, as well as recirculating immune cells (predominantly antigen-presenting cells and lymphocytes). Additionally, the tissue of origin and its physiological state have a major impact on the composition of the lymph [5]. For example, lymph from the liver is particularly rich in protein, whereas lymph from the intestinal mucosa, in particular following a meal, is rich in fat-transporting chylomicrons, giving it a milky appearance. It is therefore called chyle (from the Greek word χυλός (chylos), “juice”). This is due to chylomicron uptake by lacteals, blunt-ended lymphatic capillaries present in the intestinal villi that are connected to submucosal lymphatic precollectors and collectors. These “milky veins” were discovered by the Italian physician Gaspare Aselli in 1622 during the vivisection of a dog that had been richly fed just prior to the operation, leading to one of the first anatomic descriptions of lymphatic vessels [6].

In contrast to thin-walled initial lymphatics, collecting lymphatic vessels show a structure anatomically similar to that of small veins, including a continuous basement membrane, continuous zipper-like junctions, and smooth muscle cell coverage. Lymphatic smooth muscle cells are crucial for the transport of the lymph against a pressure gradient and act through rhythmic pumping contractions, whereas valves ensure flow direction. The largest collectors, the lymphatic ducts, are histologically similar to medium-sized veins.

The thoracic duct, the largest of all lymphatic vessels in the body, processes together with the aorta in front of the vertebral bodies through the *Hiatus aorticus* of the diaphragm and connects via an arc-shaped course to the left venous angle (Figure 1B). The right lymphatic duct, which collects lymph from the right upper body, connects to the right venous angle, formed by the right subclavian vein and right internal jugular vein. At the sites of lymphatic-venous anastomosis, lympho-venous valves prevent backflow of blood into the lymphatic system. Together, the human thoracic and right lymphatic ducts carry two to three liters of lymph back into the venous system every day [7].

Disturbances within the lymphatic system may result in lymphedema [8]. This occurs when there is an imbalance between the leakage of fluid from the blood circulation (relatively too large) and lymphatic drainage (relatively too small). Increased leakage from blood vessels is common, e.g. during acute inflammation, and may result in transient tissue swelling if the drainage capacity of the lymphatic system is exceeded. On the other hand, lymphatic vasculature malfunction is often associated with chronic lymphedema [9]. In the case of primary lymphedema, mutations in crucial lymphatic genes such as vascular endothelial growth factor receptor 3 (VEGFR-3) or forkhead box c2 (FOXC2) result in lymphatic hypoplasia and impaired function (Milroy disease, Meige disease) [10]. Secondary lymphedema occurs as a consequence of trauma, surgery (such as LN resection after breast cancer), infection (as in filariasis), or radiation therapy [10,11].

LNs are circular to kidney-shaped “filter stations” and secondary lymphatic organs intercalated in the lymphatic vascular tree (Figure 1B). Human LNs are typically sized in the mm–cm range and are surrounded by a fibrous tissue capsule with extensions (trabeculae) that protrude radially inwards, similar to wheel spokes. At the convex side of the organ, numerous afferent lymphatic vessels pass the capsule into the LN. At the hilum, situated at the concave side, usually a single draining lymphatic vessel exits the node. This is also where blood vessels enter and exit the LN. Mouse LNs, apart from being smaller, show a similar but more simple anatomy, without trabeculae.

Peripheral LNs typically receive afferent lymph from a specific body region, e.g., from one extremity or from an organ such as the stomach, where they species-dependently appear as single (e.g., in the mouse) or groups of LNs (e.g., in humans). However, there are also well-documented cases of “LN sharing”, when distinct LNs receive lymphatic output from several organs simultaneously [12]. Efferent lymphatic vessels may subsequently channel the lymph to secondary, collecting LNs before it finally pours out into the venous system.

Specific for humans and of great clinical relevance are the “Virchow’s LNs” or left supraclavicular LNs, the last LNs along the thoracic duct before the lymph enters the venous circulation. Since the thoracic duct drains the lymph of the majority of the body including the entire abdomen, enlarged and hardened Virchow’s LNs, also known as the Troisier sign, may indicate lymphogenic metastasis spreading from the abdominal cavity [13,14]. Even today, the prognosis of malignant tumors after metastasis to Virchow’s node is considered to be extremely poor [15].

The lymphatic flow through LNs is channeled via lymph sinuses (Figure 1B). Entering afferent lymphatics open up into a large, flat lymphatic space, the marginal or subcapsular sinus (SCS). The SCS expands directly under the LN capsule and is interspersed with numerous reticular tissue pillars. The resulting reticuloendothelial meshwork ensures an open lumen and acts at the same time as a sieve for the retention of large cells and emboli, e.g., thrombi, tumor cell metastases, fat cells, or debris [16]. From here, intermediate sinuses span through the cortex and insert into a labyrinth of wide-lumened medullary sinuses. These fuse at the hilum and release the lymph into the efferent lymphatic vessel. The medullary sinus lumen contains mainly lymphocytes and macrophages. Flat LECs form the inner lining of all sinuses and ensheathe fibrous tissue strands crossing the lumen to form the above-mentioned tissue pillars. The LEC layer separates the lymph from the underlying LN pulp, composed of the cortex, paracortical zone, and medullary cords. The cortex contains the B cell follicles, while the paracortex represents the T cell zone. High endothelial venules (HEVs), specialized blood vessels equipped with unusually large, cobblestone-shaped endothelial cells, can also be found here. Notably, HEVs are an important contact site between the blood circulation and the lymphatic system, allowing the diapedesis of lymphocytes from the blood directly into the LN [17]. Lastly, medullary cords are preferentially populated by plasma cells, macrophages, and plasmacytoid dendritic cells. Accumulation of macrophages in the medullary cords is prominent in lung-draining, anthracotic LNs. Due to the storage of carbon particles or soot, these LNs appear macroscopically black, which illustrates the versatile transport capabilities of the lymphatic system.

## 2. Morphological and Functional Heterogeneity of LECs

As outlined above, LECs line the lumen-facing surface of all lymphatic vessels and of the lymphatic sinuses within LNs. These cells commonly express pan-endothelial marker proteins such as PECAM-1 (CD31) and VE-cadherin (CDH5) as well as specific lymphatic markers such as podoplanin, VEGFR-3, and the transcription factor PROX-1, which is crucial for lymphatic endothelial identity [18,19]. In addition to these unifying molecular criteria, however, LECs have been recognized as a morphologically and functionally heterogenous cell population (Figure 1C). A well-described example of this heterogeneity are the LECs forming initial and collecting lymphatic vessels, which are clearly distinct. Initial LECs are irregularly shaped cells, expressing the hyaluronan receptor LYVE1, and are separated by button-like junctions. Lacking a strong, continuous basement membrane, they are equipped with anchoring filaments that connect them to the surrounding extracellular matrix (ECM) to prevent collapse in the event of high interstitial fluid pressure. Collecting LECs in contrast have a regular, elongated shape, are connected by tight zipper-like junctions, and lack LYVE1 expression [20].

Since LECs are lining the LN sinus lumen, these cells are positioned to control the access of soluble molecules and subcellular particles (including viruses) from the lymph to the LN pulp and to the conduit system. Moreover, resident LN LECs act as “stationary immune cells”, as they control lymphocyte egress from LNs via generation of a sphingosine-1-phosphate (S1P) gradient [21,22] and provide peripheral self-tolerance due to the expression and presentation of peripheral tissue self-antigens [23,24,25]. Similar to professional antigen-presenting cells, LN LECs have been reported to scavenge and (cross-) present exogenous antigens taken up from the lymph [26]. Interestingly, they also store foreign antigens (e.g., virus-derived antigen) over extended periods of time, thereby contributing to the maintenance of immunological memory [27]. At the same time, LN LECs generally lack co-stimulatory molecules, whereas they constitutively express programmed death ligand 1 (PD-L1) and other immune-inhibitory molecules [28]. Deletion of LEC-expressed PD-L1 resulted in improved CD8+ T cell responses to tumor antigens in tumor-draining LNs [29]. Thus, antigen-presenting LN LECs may inhibit T cell immunity by default.

Notably, LECs lining the ceiling (cLECs) and the floor (fLECs) of the SCS exhibit remarkable molecular differences. For example, cLECs express the atypical chemokine receptor (ACKR4), which has been shown to scavenge dendritic cell- and T cell-attracting chemokines, simultaneously displaying no or only low levels of LYVE1. In contrast, fLECs express high levels of LYVE1 but are negative for ACKR4. Further studies identified mucosal vascular addressin cell adhesion molecule 1 (MADCAM1) and integrin subunit alpha 2b (ITGA2B) as additional markers of fLECs [23,30,31]. Interestingly, the distinct molecular phenotypes of cLECs and fLECs resemble those of collector (LYVE1-negative) and initial (LYVE1-positive) lymphatic vessel LECs. Presumably, these characteristics enable these LEC subsets to support comparable functions, such as transmigration of antigen-presenting cells from the SCS to the LN cortex in case of fLECs or the formation of a tight barrier towards the surrounding tissue in case of cLECs [32].

Using single-cell RNA sequencing (scRNA-seq), several studies comprehensively mapped the differences between cLECs and fLECs. In both mice and human subjects, these studies not only unearthed novel markers and potential functions, but also defined additional LN LEC subpopulations, most importantly LECs residing in medullary sinus (mLECs) [32,33,34]. ScRNA-seq also made it possible to document possible dynamic responses of LN LECs to pathological conditions (e.g., skin inflammation and melanoma growth), providing a broad overview of LN LEC states and molecular adaptability at the single cell level [33,35,36].

## 3. Lymphatic Remodeling in Pathological Conditions

Growth of the lymphatic system is mainly restricted to the embryonic phase. However, in pathological conditions, such as chronic inflammation, impaired wound healing, and cancer, de novo lymphangiogenesis and dilation of existing lymphatic vessels can be induced and are suspected to be important, but often overlooked, components in pathology. Indeed, solid tumor growth is frequently associated with expansion of lymphatic vessels, particularly at the tumor edges, and the extent of cancer lymphangiogenesis strongly correlates with negative outcome in a vast range of cancer types. While tumor-associated lymphatic vessels may affect tumor progression in multiple ways (reviewed in [3]), one important factor underlying this correlation is lymphogenic metastasis, occurring when malignant cells emerge from a primary tumor, gain access to lymphatic vessels, are carried away with the lymph flow, and form daughter tumors elsewhere (often in regional LNs). Correspondingly, the occurrence of lymphogenic metastasis correlates with lymphatic vessel density within and around the tumor tissue, expression of lymphangiogenic factors such as vascular endothelial growth factor C (VEGF-C), and with lymphatic invasion [3,37]. However, lymphatic metastasis can also be observed in tumors that do not induce significant lymphangiogenesis, suggesting that co-option of pre-existing lymphatic vessels is sufficient for this process. In the diagnosis and treatment of malignant tumors, sentinel LNs are of practical and clinical importance. These are the LNs that are the first to receive lymph from the tumor area. A positive sentinel LN status is a poor prognostic indicator in many cancer types and is associated with an elevated probability of further metastases.

Many types of cancer, including melanoma, breast cancer, squamous cell carcinoma of the head and neck, pancreatic cancer and cervical cancer, tend to disseminate through the lymphatic system. There is still no complete explanation why these tumors tend to metastasize to LNs, while others intravasate directly into blood vessels and reach distal sites via the blood stream. The concept of the “pre-metastatic niche” (PMN) could potentially explain some of this unpredictable behavior [38]. According to this concept, factors secreted by tumor cells educate the microenvironment where metastases later develop.

Interestingly, LNs undergo dramatic remodeling even before the occurrence of LN metastasis, including a general increase in LN size, follicular and paracortical hyperplasia (involving recruitment of immune-inhibitory myeloid cells and regulatory T cells), expansion and phenotypic changes of LN LECs [3]. Currently, the main events and signals responsible for this process are not entirely clear. Tumor microenvironment-derived secreted proteins, such as growth factors and cytokines (e.g., VEGFs [39,40]), have been demonstrated to contribute to pre-metastatic LN remodeling. Additionally, accumulating data from animal studies and cancer patients strongly indicate that tumor cell-derived extracellular vesicles (EVs) are a crucial component involved in this process.

## 4. Lymphatics as Transport Route for Tumor Cell-Derived EVs to Draining LNs

### 4.1. Principles of Lymphatic Transport

Interstitial fluid forms by blood capillary filtration, percolates through gaps in the extracellular space at rates of 0.1–1.0 mm/s, and subsequently enters initial lymphatic vessels through the characteristic gaps (i.e., button-like junctions) between LECs [41,42]. With this fluid, lymphatic vessels take up tissue-derived solutes, metabolites, antigens, proteins, as well as larger particulates and protein complexes, which are efficiently transported by lymphatic vessels and accumulate in draining LNs [43]. Even migrating cells, such as antigen-presenting dendritic cells (DCs), are able to squeeze through the wide flap junctions formed by initial LECs, indicating that lymphatic vessels transport a variety of cargoes over a large size span.

Numerous techniques have been used in pre-clinical animal models to experimentally assess the efficiency of lymphatic transport of solutes as well as molecular or particulate cargoes. For example, fluorescently tagged or radiolabeled tracers such as dextrans, liposomes, and nanospheres have been used to measure the lymphatic drainage capacity in vivo after injection into the interstitium [44,45,46,47]. Based on such studies, the current notion is that substances with a molecular weight below 16 kD have direct access to blood vessels and are not specifically drained by the lymphatic system [44], whereas 40 and 500 kDa dextrans (corresponding to a hydrodynamic diameter of 10 nm and 30 nm, respectively) are efficiently transported by lymphatics. In contrast, several studies using experimental tracers such as polystyrene spheres and liposomes with diameters of 50 nm and above concluded that objects in this size range are not efficiently transported by lymphatic vessels [46,48]. However, while an inverse correlation between diameter and transport efficiency was noted, there is no evidence of a definite upper size cutoff for lymphatic uptake other than the distance between neighboring button-like junctions, which is several µm. Thus, entry of particulate cargoes in the size range of 50 nm to approximately 3 µm from the surrounding tissue into the lymphatic lumen might primarily depend on their mobility within the interstitium.

Importantly, cargo size is not the only, and maybe not even the most important, determinant of lymphatic transport efficiency. Other physical traits, such as surface charge or shape of nanoparticles, appear to play a similarly crucial role in interstitial transport and lymphatic uptake. For example, positively charged nanoparticles exhibit a higher capacity to cross blood vessel endothelia compared to their neutral or anionic counterparts [49,50,51]. On the other hand, neutral nanoparticles diffuse faster and distribute more homogeneously within the interstitial space than cationic and anionic particles, because the latter form aggregates with negatively charged (for example, hyaluronan) or positively charged (for example, collagen) matrix molecules [52,53]. Near-neutral surface charge was also found to be optimal for lymphatic uptake and transport of nanoparticles [54]. As far as the particle shape is concerned, studies have shown that macromolecules with linear, semi-flexible configurations diffuse more efficiently in the interstitial matrix than comparably sized, rigid spherical particles [55,56]. Moreover, deformability conceivably facilitates uptake, especially of large cargoes, but its impact has not been studied experimentally yet.

Taken together, lymphatic vessels are able to take up and transport a wide range of molecular and particulate cargoes, but the precise physical determinants of efficient interstitial mobility and lymphatic uptake are not entirely understood. Furthermore, it is important to keep in mind that most of the aforementioned studies employed artificial tracers and may thus not be fully applicable to natural lymphatic cargoes, which differ not only in physical but also in biochemical properties, such as the presence of specific surface molecules.

### 4.2. In Vivo Evidence for Lymphatic Transport of Tumor Cell-Derived EVs

EVs are defined as natural, subcellular microparticles surrounded by a double layer lipid membrane. Originally, EVs were broadly categorized into exosomes, microvesicles or ectosomes and apoptotic bodies. Exosomes are derived from endosomal membranes through inward budding, forming multivesicular bodies that can fuse with the plasma membrane to release their vesicular content, whereas microvesicles directly bud from the plasma membrane of the donor cells. In addition, several other EV subsets have been described more recently, such as migrasomes and amphisomes [57]. However, due to major difficulties to distinguish experimentally between those origin-based EV classes, characteristics of EVs that can be determined experimentally, such as the diameter, are currently preferred for EV classification [58]. For example, EVs with a diameter of up to 200 nm are now commonly designated as small EVs (sEVs), whereas those with diameters above 200 nm are designated as large EVs (lEVs), irrespective of their membrane of origin.

EVs contain donor cell-derived proteins, nucleic acid species, and other biomolecules. Consequently, EVs have been suggested to act as intercellular signaling entities that promote molecular communication between different cell types and organs within the body [57,59,60,61] by delivering bioactive cargo over short and long distances. The mechanisms by which the different EV subtypes are secreted by donor cells and are taken up by recipient cells have been discussed in great detail recently [57,61], but the processes underlying EV transport between donor and recipient cells are less well understood. Short-distance communication clearly takes place locally between neighboring cells or through channels in the interstitium, whereas long-distance communication requires a corresponding transport system to reach distal sites. Many studies focused on the blood circulation for EV transport [60]. However, given the interstitial fluid dynamics and lymphatic transport capacity discussed above, it is conceivable that the lymphatic vascular system is a primary route for regional and distal EV distribution [59]. Once released into the interstitium, EVs are likely taken up by lymphatic vessels and transported to draining LNs. From there, EV access to the blood circulation may either occur through HEVs or through the junction between the lymphatic ducts and the venous circulation [59].

Most in vivo evidence for lymphatic transport of EVs stems from studies with tumor cell-derived EVs in mouse models, typically using EVs produced in vitro, labeled with fluorescent dyes or other tracers, and subsequently injected interstitially. Hood et al. demonstrated that B16-F10 melanoma EVs labeled with the lipophilic, far-red fluorescent dye DiR that were injected into the footpad of C57/Bl6 mice could be detected in the inguinal LN after 48 hours [62]. Furthermore, by using in vivo near-infrared imaging of the tail of Balb/C mice, Srinivasan et al. were able to show that tumor EVs labeled by IRDye800 conjugated to EV proteins were transported from the periphery to the draining LN within minutes after interstitial injection into the tail [63]. Garcia-Silva et al. confirmed these findings and showed that DiD-labeled EVs derived from several metastatic melanoma cell lines could be detected in both popliteal and inguinal LNs [64] after footpad injection, whereas Leary et al. reported the localization of B16-F10-derived EVs predominantly in the popliteal LN after subcutaneous injection at the dorsal aspect of the hind paw [65]. The difference between these findings can probably be explained by experimental parameters such as EV or label dosage, precise injection site, and timepoint of the readout. Of note, EVs derived from other cancer models appear to behave similarly; for example, bladder cancer-derived EVs [66] and colorectal carcinoma-derived EVs [67].

The above-mentioned results were obtained by bulk injection of extracorporeally generated EVs that were labeled with fluorescent dyes intercalating into membranes or covalently bound to EV proteins, resulting in several limitations that need to be considered. For example, many commonly used lipophilic dyes (such as PKH, DiD or DiR), are not covalently bound to the EV membrane and can spontaneously “bleed” from labeled EVs. Furthermore, these dyes tend to form micelles in aqueous solutions that may have similar sizes as EVs. In addition, there is no consensus on physiologically relevant doses of injected EVs. To overcome these limitations, genetic tools have been developed to label and track tumor EVs directly in vivo. Pucci et al. engineered B16-F10 melanoma cells to express luciferase fused to a protein scaffold naturally enriched in EVs, resulting in EV-associated luciferase activity. Implanting these tumor cells into mice, they confirmed that EVs were distributed through the lymphatic system and accumulated in tumor-draining LNs [68]. Similarly, B16-F10 cells expressing palmitoylated (membrane-tethered) GFP released GFP-positive EVs after tumor formation in vivo, and these tumor-derived EVs were found to be taken up by LN LECs [65]. Together, these studies clearly demonstrate that in murine models, cancer cell-derived EVs spread from the interstitium through the lymphatic system and accumulate in LNs.

### 4.3. Tumor EVs in the Lymph of Cancer Patients

Whereas animal models have been instrumental to track in vivo behavior of EVs after injection or release from implanted tumor cells, confirming their relevance and translatability to human subjects is challenging. Of note, cancer patients often present with malignancies that developed over years and are already progressed. Correspondingly, tumor cell EV release and biodistribution may be very different compared to the comparably acute animal models, and EV-mediated PMN formation may be of limited relevance. Nonetheless, several studies provided evidence for tumor-derived EVs in peripheral blood [69], urine [70], cerebrospinal fluid [71], and lymph-rich wound exudate of cancer patients [72,73]. Notably, characterization of postoperative wound exudate and plasma of metastatic melanoma patients after lymphadenectomy revealed that the number and size of EVs were increased in the wound exudate and that exudate EVs were highly enriched for tumor cell-related proteomic patterns compared to the plasma [72,73]. Under these particular circumstances though, there is a risk for cross-contamination, since exudate samples taken after lymphadenectomy may contain plasma-derived components [73]. Yet, Maus et al. freshly isolated human afferent lymph draining directly from primary cutaneous melanoma at the time of initial sentinel LN biopsy, and the authors were able to identify EVs in the lymphatic fluid of the patients [74]. Thus, as in animal models, there is solid evidence that tumor cell-secreted EVs are predominantly transported through the lymphatic system in melanoma patients.

## 5. Fate and Function of Tumor Cell-Derived EVs in the Lymphatic System

### 5.1. EV Recipient Cells in the LN Microenvironment

Several studies found that at least in the case of tumor cell-derived EVs, LNs act as filters that retain and accumulate EVs [62,64,65,68], suggesting that EVs are scavenged by or interact with specific cell populations within LNs. Indeed, some tumor EV recipient cell types have been identified using in vivo models (Figure 2). Importantly, LN LECs have been found to be major recipients of cancer cell-derived EVs, whereas minimal EV uptake was detectable in other LN stroma cells (blood endothelial cells (BECs), fibroblastic reticular cells (FRCs)), DCs, or lymphocytes [64,65,73,75]. Additionally, uptake was strongest among fLECs compared to cLECs and mLECs [65]. A second prominent cell population taking up tumor EVs from the afferent lymph is LN-resident macrophages. In the case of melanoma-derived EVs, these were, in particular, CD169+ macrophages populating the lymphatic sinuses [64,65,68,76], whereas colorectal carcinoma-derived EVs were predominantly taken up by F4/80+ macrophages [67]. Thus, recipient cell types in the LN microenvironment might differ depending on the identity of the EV donor cell.

How exactly tumor EVs interact with their recipient cells in the LN microenvironment is not fully understood. Multiple cell type-specific and non-specific interaction and uptake mechanisms of EVs have been identified previously [58,60,61,77,78]. In the case of LN LECs and sinusoidal macrophages, their location in direct exposure to the afferent lymph is probably relevant for the uptake of EVs drained from upstream tissues. In addition, several studies provided evidence for potential specific uptake mechanisms. For example, B16-F10-derived EVs carry multiple integrins, and the blockade of alpha-v and beta-1 integrins reduced EV uptake by LN LECs in vivo [64,65]. Consistently, deletion of LEC-expressed VCAM-1, a ligand of alpha-4/beta-1 and alpha-9/beta-1 integrins present in B16-F10-derived EVs, reduced EV uptake by LN LECs [65]. However, in neither case was EV uptake abolished completely, suggesting that multiple molecular interactions may be involved in the binding and uptake of melanoma EVs by LN LECs. Furthermore, blockade of the aforementioned integrins did not affect the uptake of EVs by LN macrophages. Thus, other receptors or ligands must play a role in the interaction of melanoma EVs and macrophages; for example CD169 itself, which binds to alpha-2,3-linked sialic acid glycoconjugates which may be present on the EV surface [76]. Nonetheless, it is conceivable that multiple additional pathways mediating interactions and uptake of EVs exist, some of which could be specific for certain cell or EV subsets. A detailed exploration of those pathways, for instance, using a recently developed method called TurboID-EV [79], might pave the way to specific therapeutic targeting of those interactions.

### 5.2. EV-Mediated Effects on LN PMN Formation

As discussed above, the PMN in tumor-draining LNs is characterized by morphological, cellular, and molecular changes, including swelling of the LN, proliferation of LECs and expansion of lymphatic sinuses, infiltration of immature and immune-inhibitory leukocyte populations, etc. Together, these changes enhance lymph flow through (and thereby the transport of metastatic cells to) the node, expand immune-regulatory and tolerogenic cell types (including LECs, regulatory T cells, myeloid-derived suppressor cells, and immature DCs) and alter tumor antigen access to the LN cortex and paracortex, ultimately facilitating lymphatic metastasis. Notably, tumor cell-derived EVs have been found to promote PMN formation not only at distant sites, but also in tumor-draining LNs [80]. In fact, interstitial injection of tumor cell-derived EVs is sufficient to evoke many of the PMN hallmarks in draining LNs, including a marked expansion of the LN as well as LEC proliferation [62,64,65,67].

Using animal models, several mechanisms by which EVs may elicit these effects have been described (Figure 3). In the case of melanoma, EVs enriched with nerve growth factor receptor (NGFR) were found to facilitate lymphangiogenesis and metastasis through the activation of ERK and NF-κB pathways in LECs, resulting in proliferation and upregulation of adhesion molecules [64]. Additionally, single-cell RNA sequencing indicated that melanoma EVs can induce the expression of lymphotoxin β (Ltb) in LECs, a cytokine crucially involved in LN formation and homeostasis [65]. In a colorectal carcinoma model, uptake of tumor cell-derived EVs by LN-resident macrophages resulted in induction and secretion of the lymphangiogenic growth factor VEGF-C [67]. Moreover, tumor cell-derived EVs have been shown to stimulate lymphangiogenesis through the transfer of various RNA species such as miR-1246 or miR-221-3p [81,82] and the lncRNA LNMAT2 [83], RNA-binding proteins stabilizing PROX-1 expression in LECs [84], and direct association of EVs with VEGF-C protein [85]. While there is currently no evidence that any of those mechanisms contribute to LN lymphangiogenesis, these findings highlight the complex roles of EVs in promoting lymphangiogenesis and lymphatic metastasis, suggesting a range of potential new therapeutic targets to prevent cancer progression.

### 5.3. EV-Mediated Effects on Tumor Immunity in Tumor-Draining LNs

Tumor-draining LNs are not only relevant as sites for lymphatic dissemination, but are also crucial for tumor immunity and tumor immune evasion [3,86]. Tumor-derived EVs play a multifaceted role in shaping tumor immunity within the LNs by transporting bioactive molecules that can either promote immunosuppression or stimulate immune activation (Figure 4). For example, tumor-derived EVs have been reported to display PD-L1 on their surface, enabling them to directly inhibit T cell responses through engagement of PD-1 [87]. Furthermore, EVs extracted from human melanoma cell lines, the serum of melanoma patients [88], and prostate cancer cell lines [89] contain FasL, which induces apoptosis in CD8^+^ T cells. Additional research confirmed that melanoma EVs isolated from the plasma of these patients consistently carry FasL and TRAIL, with FasL-blocking antibodies partially reducing the apoptosis of T cells induced by these EVs in vitro [90].

Tumor cell-derived EVs have also been described to impair T cell-mediated immunity indirectly. For example, chronic lymphocytic leukemia-derived EVs induced PD-L1 expression in monocytes [91], whereas EVs isolated from the lymph of primary cutaneous melanoma patients contained S100A9, an inhibitor of DC maturation, which is crucial for immune response initiation [92]. Interestingly, tumor cell-derived EVs are rich in tumor antigens, and could, upon uptake by antigen-presenting cells, prime tumor-specific T cells. However, at least in the case of melanoma, EVs-mediated transfer of tumor antigens to LN LECs and subsequent presentation on MHC-I rather inhibited the response of CD8+ T cells, most likely due to the inherent tolerogenic capacity of LN LECs [65]. EV interactions with LN LECs were also shown to recruit granulocytes to the LN microenvironment, which may impair tumor immunity through NETosis-mediated T cell inhibition [93]. On the other hand, scavenging of melanoma EVs from the afferent lymph by CD169+ sinusoidal macrophages prevented their interaction with B cells and the initiation of tumor-promoting humoral immunity [68].

The heterogeneous nature of EVs contributes to their varied effects on tumor immunity. Some studies suggest that, under specific therapeutic conditions, tumor-derived EVs might stimulate host immunity [94]. This underscores the need for further research to clarify the mechanisms by which EVs influence tumor immunity in LNs and explore potential therapeutic targets and clinical applications.

## 6. EV—Lymphatic System Interaction as a Therapeutic Target

As discussed above, tumor cell-derived EVs promote metastasis and suppress anti-tumor immunity in multiple ways. Consequently, targeting the EV-mediated crosstalk between cancer and the lymphatic system may present new avenues for potential therapies. This could be achieved by disturbing EV biogenesis, preventing the entrance of EVs into the lymphatic network, or blocking their interaction with recipient cells in the LN microenvironment [95,96,97,98]. For instance, Nishida-Aoki et al. observed significantly lower metastasis in the lungs, LNs, and thoracic cavity upon anti-CD9 and anti-CD63 antibody treatment due to macrophage-mediated EV clearance, even though primary tumor growth was not affected [99]. In a study by Poggie et al., the suppressive effect of EV-associated PD-L1 on T cell activity in the draining LN could be eliminated by the disruption of EV biogenesis through the deletion of NSMASE2 and RAB27A genes [100]. The absence of PD-L1+ EVs significantly increased T cell activity, the effect of immune checkpoint therapy, and the overall survival in TRAMP-C2 and MC38 tumor-bearing mice [100]. Packaging the exosome release inhibitor GW4869 and ferroptosis inducer Fe^+3^ in an amphiphilic hyaluronic acid assembly, Wang et al. successfully depleted B16-F10-derived EV secretion, enhancing cytotoxic T cell activation and memory T cell formation [101]. However, the above-mentioned strategies lack specificity and/or may not be translatable into therapeutic approaches. This warrants further studies to explore tumor-specific vulnerabilities, such as EV uptake receptors or effector pathways triggered in EV recipient cell types to identify suitable molecular targets.

## 7. EVs as Biomarkers for Cancer and LN Metastasis

Tumor-derived EVs are not only of clinical interest as therapeutic targets, but also as a potential source of diagnostic and prognostic biomarkers. Especially plasma-derived EVs can easily be sampled through liquid biopsy and have been proven to hold diagnostic and predictive value in a range of cancer types [102,103,104]. In contrast, as of now, lymph-borne EVs have been less commonly explored in biomarker research, even though several studies using lymphatic exudate collected at sites of lymphadenectomy in melanoma and breast cancer patients found a superior source of EV-associated molecular information with high predictive value compared to plasma [72,73,105]. Unfortunately, sampling of lymphatic exudate is only applicable to patients undergoing lymphadenectomy, and other approaches to collect tumor-draining lymph are very challenging to implement in current clinical routines.

Beyond natural EV biomarkers, synthetic nanoparticles (or engineered EVs) might offer an additional approach for the detection of cancer and LN metastasis. For instance, Han et al. introduced silicon nanoparticle-modified EV probes as tracers to distinguishing normal from metastatic sentinel LNs, increasing the sensitivity and decreasing the wait time for illumination compared to current tracers such as indocyanine green (ICG) [106]. Clearly, EV-based methods are emerging as alternatives for detecting cancer and LN metastasis and continue to advance the available “toolbox” of diagnostic techniques.

## 8. EVs as Therapeutic Vectors

The ability of EVs to carry molecular cargo to specific tissues and cell types has excited scientists [107]. For potential therapeutic applications, EVs can be sourced from various biological fluids, such as serum and milk, providing a natural background. Conditioned cell culture media offers a controlled environment for bulk EV production and makes engineering EVs with specific therapeutic cargo possible during production. Another alternative method is creating EVs through serial extrusion of whole cells, allowing usage of natural biological cargo and lipids of the source cell, and making the homogenization and customization of vesicle properties possible. Each source presents unique advantages towards scalability, customizability, and biocompatibility for effective drug delivery applications. EVs derived from multiple cell types have been adapted, and synthetic approaches are being developed to combine the advantages of EVs and nanoparticles [108]. The lymphatic system gained great attention as a delivery path of drugs, especially to treat in-transit and LN metastasis, because it concentrates the cargo in the lymphatic lumen and the LN microenvironment while minimizing side effects [109,110,111]. In addition, lymphatic application may be optimal for EV-based immunotherapeutics. The use of EVs for immunotherapies dates back to 2005, when the first phase I clinical trial by Escudier et al. for the application of DC-derived EVs for vaccination against metastatic melanoma was published [112]. Despite mixed results, the study proved that large-scale production of EVs and administration to patients is feasible. Subsequently, further studies reported the use of macrophage-derived EVs for immunotherapy. For example, Cheng et al. utilized M1 polarized macrophage-derived EVs to develop a tumor antigen-specific melanoma vaccine [113]. M1 macrophage-derived EVs accumulated in draining LN and were taken up by macrophages and DCs, eliciting a Th1 response [113]. Furthermore, Wang et al. generated macrophage–tumor hybrid cells (aMT-cells) by introducing tumor cell nuclei into activated M1 macrophages, which secreted chimeric EVs (aMT-exos) [114]. aMT-exos accumulated in LNs, activating T cells through antigen presentation and direct interaction. Combining aMT-exos with anti-PD-1 therapy further improved survival in metastatic and postsurgical tumor recurrence models [114].

While the above-mentioned studies used EVs directly collected from donor cells for therapeutic purposes, another approach is to functionalize EVs by loading them with specific payload molecules to elicit the desired effects within the lymphatic system or in LNs. Such therapeutic payloads may be peptides (often for vaccination purposes) or whole proteins, but also nucleic acid species, such as mRNA or DNA species, have been incorporated into EV-based therapeutics successfully. For example, the Mok group loaded serum-derived EVs with Trp2 peptides and monophosphoryl lipid a (MPLA) (EXO-MPLA-TRP2) to target macrophages and DCs in the draining LN [115]. The authors were able to show that the uptake of EXO-MPLA-TRP2 promoted the release of pro-inflammatory cytokines TNF-α and IL-6 [115]. In another study, Phung et al. functionalized DC-derived EVs with anti-CTLA-4 antibody, MHC molecules, and ovalbumin peptide (EXO-OVA-mAb) via lipid-anchoring [116]. EXO-OVA-mAb induced T cell activation in the tumor-draining LN and significantly suppressed tumor growth [116]. An EV hybrid lipid nanovesicle-based cancer vaccine (Lipo@HEV) was formulated by Tong et al. [117], combining tumor cell-derived EVs and *Akkermansia muciniphila* outer membrane vesicles (Akk-OMV), which penetrate into LNs to initiate DC maturation and activation of cytotoxic T cells [117]. In addition, Lipo@HEV could be loaded with plasmid DNA to enable a gene therapy-mediated PD-L1 blockade [117]. Another study utilized engineered EVs with albumin binding domains (ABDs) to extend the circulation time in the body through surface display of ABDs on tetraspanins (CD63, CD9, and CD81) or EV-sorting domains (Lamp2B) [118]. Engineered EVs exhibited strong binding to human and mouse serum albumin, showed significantly increased circulation time in mice after various administration routes, and accumulated in LNs and solid tumors, probably due to their prolonged circulation time [118]. Further advancements in natural and synthetic EV research in combination with lymphatic application will undoubtedly open the doors for novel therapies, not only to target cancer progression and tumor immunity, but potentially also to regulate immune responses in a wide range of diseases, including autoimmunity and chronic inflammation.

## 9. Conclusions and Open Questions

The lymphatic system not only drains interstitial fluid from the entire body periphery, but also solutes and particulate cargoes, depending on physical and biochemical properties such as size, surface charge, surface molecules, etc. Experimental and clinical evidence strongly indicate that tissue-derived EVs, including EVs released by solid tumors, are efficiently taken up by initial lymphatic vessels and transported to draining LNs. In fact, given that interstitial flow is directed away from blood vessels and towards lymphatics, it is conceivable that the lymphatic system represents the primary route for EV dissemination away from the tissue (or tumor) of origin. However, it is currently unknown whether EV uptake from the interstitium into the lumen of lymphatic vessels is solely mediated by flow, or if specific, cellular transport mechanisms are involved.

Importantly, the majority of EVs appears to be retained in draining LNs due to physical entrapment in small caliber lymphatic sinuses or interactions with macrophages, LECs, and potentially other cell types surveilling the lymph. Consequently, these cells are major recipients of EVs and their molecular cargoes, with potential consequences for their phenotype and function. In case of tumor cell-derived EVs, this type of cell-to-cell communication may contribute to tumor progression (through PMN formation), tumor immune evasion, etc., suggesting that therapeutic interference with the uptake of tumor EVs by their recipient cells or other downstream events could be an entirely new strategy in the portfolio of cancer therapeutics. However, our understanding of the underlying molecular and cellular events is still in its infancy, which is at least in part due to inherent challenges and a lack of rigorous methods to model and study such processes in vivo. Most importantly, we propose that future research efforts should be directed towards a deeper molecular and functional characterization of subpopulations of tumor-derived EVs, together with a precise mapping of their biodistribution, recipient cells, and in vivo uptake mechanisms. Thereby, pathologic EV subsets and new therapeutic targets, such as EV receptors expressed by recipient cells in tumor-draining LNs, may be discovered. Blockade of such receptors could be a promising approach to synergize with other types of cancer therapy, such as immunotherapy, particularly in unresectable disease or in the neo-adjuvant setting. At the same time, we need to deepen our understanding of the principles and functions of “natural” EV-mediated communication through the lymphatic system under physiological conditions to avoid potentially detrimental adverse effects of EV-targeting therapies and to inspire the design of EV-based drugs, promising to be versatile tools not only to target malignant cells within the lymphatic system, but also to molecularly “re-program” selected target cell types in the LN microenvironment with unprecedented specificity and efficiency. For example, specific EV-mediated delivery of immune-stimulatory molecules in combination with tumor-associated antigens to antigen-presenting cells in tumor-draining LNs might synergize with checkpoint blockade in cancer patients, while autoantigen delivery to tolerogenic cell types in LNs, such as LECs, might be beneficial in the context of autoimmune diseases. Witnessing a highly dynamic research community, combined with major technical advances in areas such as EV subfractionation as well as ultra-sensitive and -resolving omics platforms that are pushing our scientific horizon from “single cell” towards “single EV” analyses, we are convinced that these and other key questions in the field can be resolved and that EV-targeting as well as EV-based drugs may become a clinical reality.

## Figures and Tables

**Figure 1 cancers-16-04039-f001:**
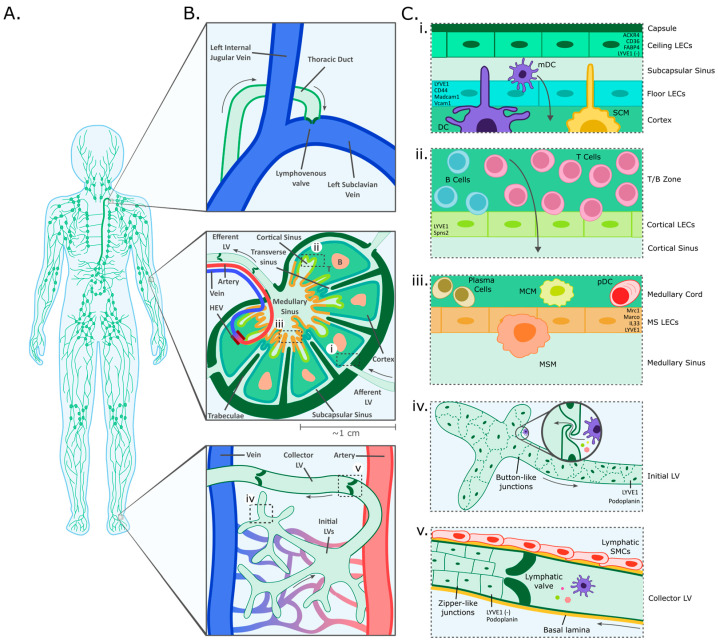
Overview of the lymphatic vascular system. (**A**) Schematic representation of the lymphatic vascular system. (**B**) Enlarged schematic figures highlight contact points between the systemic circulation and the lymphatic system (from top to bottom): the lymphovenous junction between the subclavian vein and the thoracic duct; the structure of LNs, secondary lymphoid organs comprising afferent and efferent lymphatic vessels (LV), lymphatic sinuses, B cell follicles (B) and the T cell zone (T), as well as blood vessels that include high endothelial venules (HEV); the initial lymphatic network present in virtually all tissues and draining interstitial fluid that forms through filtration from blood capillaries. (**C**) Selected regions from (**B**) as higher detailed schematics illustrating LEC subtypes, structure, and marker gene expression: (**i**) The subcapsular sinus is lined by ceiling LECs (cLECs) and floor LECs (fLECs) which differ in marker gene expression and closely interact with subcapsular sinus macrophages (SCM) and dendritic cells (DCs). (**ii**) (Para-) cortical sinuses are exit routes for recirculating T and B lymphocytes. (**iii**) Medullary sinuses are lined by medullary LECs (mLECs) and connect to the efferent vessel. They are populated by medullary sinus macrophages (MSM), whereas medullary cords are dominated by medullary cord macrophages (MCM), plasma cells, and plasmacytoid dendritic cells (pDCs). (**iv**) Initial lymphatic vessels are composed of irregularly shaped LECs that are connected by button-like junctions which facilitate intravasation of interstitial fluid, solutes, larger particles, and entire cells. (**v**) Collecting lymphatic vessels are composed of regularly shaped LECs connected by tight zipper junctions and supported by a basal lamina and lymphatic smooth muscle cells (SMCs).

**Figure 2 cancers-16-04039-f002:**
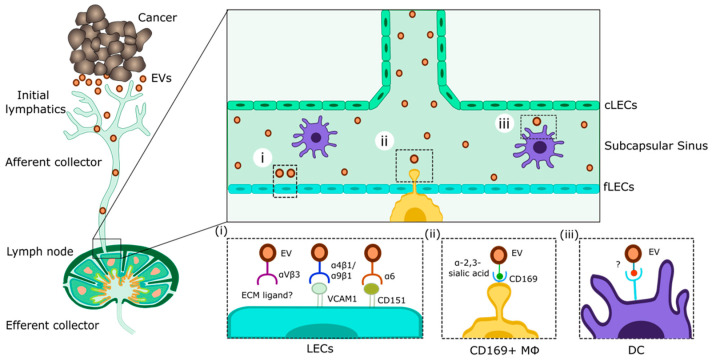
Interactions between tumor cell-derived EVs and EV recipient cells in tumor-draining LNs. EVs released by tumor cells into the interstitium are taken up by tumor-associated lymphatic vessels and transported to draining LNs. Here, EVs may interact with antigen-presenting cells such as LECs (**i**), sinusoidal macrophages (**ii**), or DCs (**iii**). The enlarged boxes indicate some of the molecular interactions between melanoma EVs and their recipient cells that have been described until today. ECM: extracellular matrix. Further explanations in the text.

**Figure 3 cancers-16-04039-f003:**
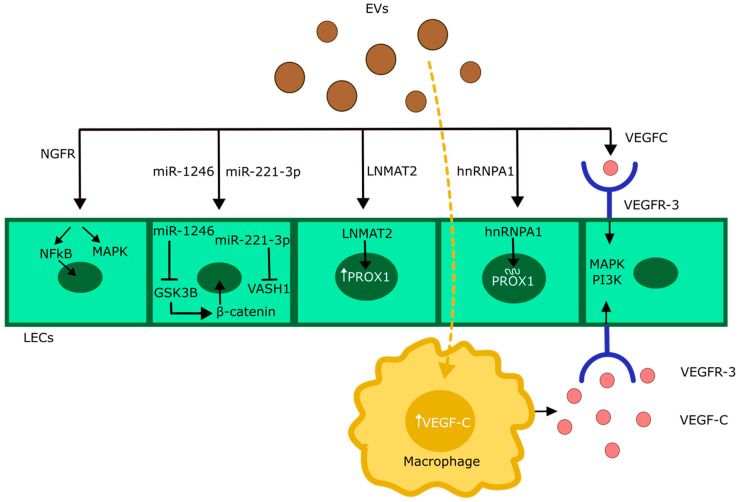
Schematic overview of EV-mediated lymphangiogenesis in pre-metastatic LNs. EVs have been shown previously to transfer a range of effector molecules that can trigger LEC expansion. Melanoma-derived EVs carrying NGFR can activate NF-κB and MAPK signaling pathways in LECs, resulting in LEC proliferation. EVs have also been shown to carry microRNAs, such as miR-1246 and miR-221-3p, which downregulate GSK3β and VASH1, leading to increased β-catenin signaling. Additionally, the lncRNA LNMAT2 and RNA-binding protein hnRNPA1 delivered by EVs to LECs help to upregulate and to stabilize the mRNA coding for the transcription factor PROX1, which is critical for LEC differentiation. Lastly, EVs may associate with the lymphangiogenic growth factor VEGF-C or induce VEGF-C expression in LN macrophages, resulting in the activation of LEC-expressed VEGFR-3 and its downstream signaling pathways. Further explanations in the text.

**Figure 4 cancers-16-04039-f004:**
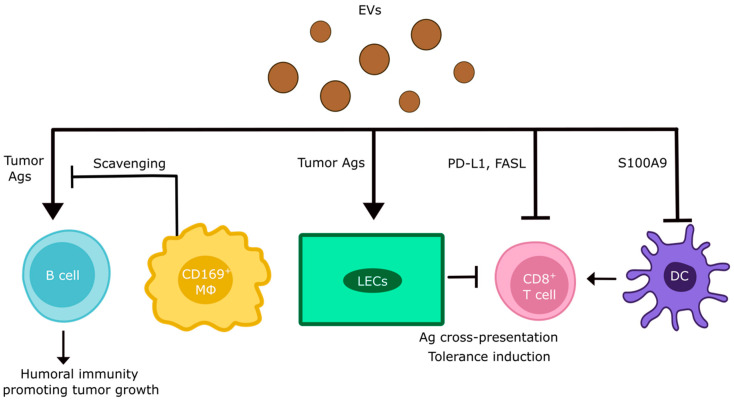
Impact of tumor cell-derived EVs on tumor immunity in draining LNs. Tumor-derived EVs may display T cell inhibitory molecules on their surface, such as PD-L1 and FasL. Furthermore, EVs have been reported to transfer tumor antigens (Ags) to tolerogenic LECs, which can cross-present them to CD8+ T cells, leading to their inhibition. Similarly, EVs reduce DC maturation and thereby inhibit T cell responses against tumor antigens. Finally, EV interactions with B cells may trigger humoral immune responses that cause inflammation and limit cellular immunity. This pathway is controlled by sinusoidal macrophages (MΦ) that scavenge tumor EVs before they reach the B cell compartment. Further explanations in the text.

## Data Availability

No new data were created or analyzed in this study.

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
