# Peer review of "The Lymphatic Vascular System in Extracellular Vesicle-Mediated Tumor Progression"

_cancers, 2024, doi:10.3390/cancers16234039_

Round 1

Reviewer 1 Report

Comments and Suggestions for Authors

In the submitted review article entitled (The lymphatic vascular system in extracellular vesicle-mediated tumor progression), the authors provide a comprehensive review of one of the interesting topics. They provide a comprehensive overview about the role of extracellular vesicles (EVs) in tumor progression. However, several issues need to be improved before acceptance for publication.

Please provide a clear illustration of lymphatic pathways in the body and their connection with systemic circulation. Firstly, lymphatic vessels originate from the intestine (mesenteric lymph node), where ingested foods are absorbed through chylomicrons. Secondly, they originate from peripheral tissues, where large particles are reabsorbed from interstitial fluid. Finally, circulating cells and pathogenic microorganisms access the lymphatic system through systemic circulation through high endothelial venules (HEV). Please try to explain this in the revised version of the manuscript, which gives the readers a clear idea of the lymphatic system. If applicable, please provide this mapping and connection between the lymphatic system and systemic circulation in the figure.

Kindly, discuss the potential sources of EVs to be used as drug delivery systems for loaded cargo. For example, it could be obtained from the biological fluids of volunteers. These biological fluids could be serum, milk, etc. Moreover, conditioned media could be used as a source of EVs. This added advantages in terms of cell engineering, where therapeutic cargo could be inherently loaded within EVs during their production. Finally, EVs could be attained from whole cells exposed to the serial process of extrusion.

Please try to add a section containing the classification of cargo based on their nature as therapeutic drug carriers. For example, therapeutic molecules (drugs, proteins, etc.) and genetic materials (mRNA, DNA, etc.).

Author Response

Point-by-point responses to the reviewer comments.

Reviewer 1:

In the submitted review article entitled (The lymphatic vascular system in extracellular vesicle-mediated tumor progression), the authors provide a comprehensive review of one of the interesting topics. They provide a comprehensive overview about the role of extracellular vesicles (EVs) in tumor progression. However, several issues need to be improved before acceptance for publication.

Please provide a clear illustration of lymphatic pathways in the body and their connection with systemic circulation. Firstly, lymphatic vessels originate from the intestine (mesenteric lymph node), where ingested foods are absorbed through chylomicrons. Secondly, they originate from peripheral tissues, where large particles are reabsorbed from interstitial fluid. Finally, circulating cells and pathogenic microorganisms access the lymphatic system through systemic circulation through high endothelial venules (HEV). Please try to explain this in the revised version of the manuscript, which gives the readers a clear idea of the lymphatic system. If applicable, please provide this mapping and connection between the lymphatic system and systemic circulation in the figure.

Response:

We thank the reviewer for this suggestion and have extensively revised Figure 1. In panel B, we now highlight the connections between the lymphatic system and the systemic circulation, including the lymphovenous junction, HEVs in lymph nodes, and initial lymphatic networks. We chose not to specifically depict intestinal lymphatics, since they represent a specialized type of an initial lymphatic network which is represented in the revised figure already.

The uptake and drainage of chylomicrons and other particles by initial lymphatics is already discussed in the main text (chapter 1, 2nd paragraph). We further emphasized the importance of HEVs as contact point between the blood circulation (and leukocytes within) and the LN parenchyma (chapter 1, last paragraph).

Kindly, discuss the potential sources of EVs to be used as drug delivery systems for loaded cargo. For example, it could be obtained from the biological fluids of volunteers. These biological fluids could be serum, milk, etc. Moreover, conditioned media could be used as a source of EVs. This added advantages in terms of cell engineering, where therapeutic cargo could be inherently loaded within EVs during their production. Finally, EVs could be attained from whole cells exposed to the serial process of extrusion.

Response:

Following the reviewer’s suggestion, we have included a discussion of different EV sources for therapeutic applications in the beginning of chapter 8.

Please try to add a section containing the classification of cargo based on their nature as therapeutic drug carriers. For example, therapeutic molecules (drugs, proteins, etc.) and genetic materials (mRNA, DNA, etc.).

Response:

We have revised chapter 8 to more clearly distinguish between unmodified EVs and those loaded with specific therapeutic molecules (peptides, proteins, nucleic acid).

Reviewer 2 Report

Comments and Suggestions for Authors

·         To contextualise the study's relevance, the manuscript should include a more extensive summary of the lymphatic vascular system's role and significance in tumour advancement.

·         The abstract might benefit from a more explicit presentation of the research objectives, notably the predominant emphasis on extracellular vesicles (EVs) in lymphatic-mediated tumour growth.

·         Some parts are extensive and might be divided into smaller subsections, particularly those addressing lymphatic transport and EV functions, to improve reading and clarity.

·         Figure 1 shows ambiguous label descriptions, notably for human and mouse lymph node designs. Label simplification and enlargement could be beneficial.

·         Some recent investigations on EV absorption pathways in the lymphatic system seem to be missing. Including references to current studies on lymphatic-specific EV transmission would improve the literature overview.

·         The methodology for tracking EV transport using animal models might be enhanced to provide more openness about experimental controls and limits.

·         Additional statistics for in vivo investigations, such as sample size, power analysis, and significance levels, would improve the study's reproducibility.

·         Ensure consistent language, as terms such as "lymphatic endothelial cells" and "LECs" are used interchangeably, potentially confusing readers unfamiliar with the concepts.

·         In sections detailing the impact of EVs on lymph node architecture, include further information about the functional consequences of these changes for tumour growth and immune responses.

·         Consider including a table summarising the various types of EVs, their sizes, and known transportation techniques to help readers visualise this information.

·         The process by which EVs stimulate LN lymphangiogenesis in pre-metastatic lymph nodes is complex; a more extensive figure legend describing each component in Figure 3 would be beneficial.

·         The study might go into greater detail about the limits of translating animal model findings to human applications.

·         When discussing EV classification, provide clarity by consistently explaining categories such as exosomes and microvesicles, which overlap in terminology.

·         While there is a brief discussion of therapeutic implications, more specific recommendations for future study approaches would be helpful.

·         The conclusion might be expanded to provide more specific insights into how this research may affect therapeutic tactics targeting the lymphatic system in cancer treatment.

Author Response

Point-by-point responses to the reviewer comments.

Reviewer 2:

  • To contextualise the study's relevance, the manuscript should include a more extensive summary of the lymphatic vascular system's role and significance in tumour advancement.

Response:

We’d like to thank the reviewer for the constructive comments! Given the specific focus of our review article, a broad discussion of the role of the lymphatic system in cancer would be beyond the scope. Nonetheless, we have amended section 3 to emphasize the role of lymphatics in tumor dissemination. For a deeper discussion of this and other, potentially tumor progression-promoting functions, the readers may refer to reference No. 3.

  • The abstract might benefit from a more explicit presentation of the research objectives, notably the predominant emphasis on extracellular vesicles (EVs) in lymphatic-mediated tumour growth.

Response:

We have revised the abstract to give further emphasis on the main topic of this review article.

  • Some parts are extensive and might be divided into smaller subsections, particularly those addressing lymphatic transport and EV functions, to improve reading and clarity.

Response:

Chapters 4 (lymphatic transport) and 5 (EV functions) are already clearly structured and broken down into 3 sub-sections each (4.1 Prinicples of lymphatic transport; 4.2 In vivo evidence for lymphatic transport of Tumor cell-derived EVs; 4.3 Tumor EVs in the lymph of cancer patients; 5.1 EV recipient cells in the LN microenvironment; 5.2 EV-mediated effects on LN PMN formation; 5.3 EV-mediated effects on tumor immunity in tumor-draining LNs), all of which are coherent and concise. We are convinced that further fractionation of the text will not improve clarity.

  • Figure 1 shows ambiguous label descriptions, notably for human and mouse lymph node designs. Label simplification and enlargement could be beneficial.

Response:

We agree that the depiction of human vs. mouse LNs was not very clear and have extensively revised Figure 1. Please also refer to our answer to the first comment by reviewer 1.

  • Some recent investigations on EV absorption pathways in the lymphatic system seem to be missing. Including references to current studies on lymphatic-specific EV transmission would improve the literature overview.

Response:

Our mansuscript includes a current discussion of specific EV uptake pathways by LECs in chapter 5.1. An additional discussion of general EV uptake pathways is beyond the scope of this manuscript. Instead, we refer the readers to excellent articles on that topic in line 397 (Mathieu et al, Nat Cell Biol 2019; van Niel et al, Nat Rev Mol Cell Biol 2020).

Whether the initial absorption of cancer EVs from the interstitium into the lumen of lymphatic vessels is mediated by passive transport or through active absorption pathways is very interesting, but to the best of our knowledge, there are no studies demonstrating such absorption pathways yet. We have highlighted this question in the revised manuscript in chapter 9.

  • The methodology for tracking EV transport using animal models might be enhanced to provide more openness about experimental controls and limits.

Response:

We thank the reviewer for this constructive comment. We have expaned section 4.2 to clarifiy the methods used to track EVs in vivo, together with their limitations.

  • Additional statistics for in vivo investigations, such as sample size, power analysis, and significance levels, would improve the study's reproducibility.

Response:

We would like to point out that this is a review article, not an experimental study. Questions of statistics and reproducibility are thus not applicable.

  • Ensure consistent language, as terms such as "lymphatic endothelial cells" and "LECs" are used interchangeably, potentially confusing readers unfamiliar with the concepts.

Response:

“LECs” is a common abbreviation for “lymphatic endothelial cells”. We introduced this abbreviation once in the first paragraph of chapter 1, and subsequently have used the abbreviation only. Furthermore, we have unified additional terms and abbreviations in the manuscript text.

  • In sections detailing the impact of EVs on lymph node architecture, include further information about the functional consequences of these changes for tumour growth and immune responses.

Response:

We apologize that the consequences of the remodeling of tumor-draining LNs on disease progression were not discussed in depth and have amended chapter 5.2 correspondingly.

  • Consider including a table summarising the various types of EVs, their sizes, and known transportation techniques to help readers visualise this information.

Response:

The reviewer is rising an extremely interesting question, namely how different EV subtypes behave in vivo and whether / how their biodistribution and target cell interaction is affected by the lymphatic vascular system. To our knowledge, very little is known about this question, and we consider it a top priority direction for future research. Consequently, we highlighted this point in chapter 9.

On the other hand, as long as their interactions with the lymphatic vascular system remain uncharted, a summary of previously described EV subtypes is beyond the scope of this review. Nonetheless, we have revised the first paragraph of chapter 4.2 to provide a deeper discussion of EV subtypes.

  • The process by which EVs stimulate LN lymphangiogenesis in pre-metastatic lymph nodes is complex; a more extensive figure legend describing each component in Figure 3 would be beneficial.

Response:

We thank the reviewer for this helpful suggestion. We have extended the legend to figure 3 to discuss the mechanisms depicted in the figure in much greater detail.

  • The study might go into greater detail about the limits of translating animal model findings to human applications.

Response:

Following the reviewer’s suggestion we have expanded section 4.3 to highlight the challenges associated with the translation of findings made in animal models to a human situation.

  • When discussing EV classification, provide clarity by consistently explaining categories such as exosomes and microvesicles, which overlap in terminology.

Response:

We have revised the first paragraph of chapter 4.2 to provide a deeper and clearer discussion of traditionally defined EV subtypes, especially exosomes and microvesicles

  • While there is a brief discussion of therapeutic implications, more specific recommendations for future study approaches would be helpful.

Response:

Following the reviewer’s suggestion we have revised the outlook chapter to more cleary indicate research directions we think are necessary in order to develop drugs to interfere with tumor EV-mediated communication between tumor cells and the draining LN microenvironment.

  • The conclusion might be expanded to provide more specific insights into how this research may affect therapeutic tactics targeting the lymphatic system in cancer treatment.

Response:

We propose that EV-inspired drugs may be very versatile tools to specifically deliver therapeutic cargoes to a range of cell types within LNs, thereby re-programming them to support (or inhibit) immune responses in a range of diseases not limited to cancer. We have revised the outlook chapter to include such strategies.

Reviewer 3 Report

Comments and Suggestions for Authors

Your informative and well-written article "The lymphatic vascular system in extracellular vesicle-mediated tumor progression" is an excellent review of the anatomy of the lymphatic system and the potential roles of EVs.

Author Response

Your informative and well-written article "The lymphatic vascular system in extracellular vesicle-mediated tumor progression" is an excellent review of the anatomy of the lymphatic system and the potential roles of EVs.

Response:

We thank the reviewer for the kind appreciation of our work and the highly motivating comment.

Round 2

Reviewer 1 Report

Comments and Suggestions for Authors

Authors address most of comments

Author Response

We thank the reviewer for the helpful suggestions!

Reviewer 2 Report

Comments and Suggestions for Authors

Most of the unnecessary cited references are not required in the manuscript. Authors need to remove references cited unnecessary like 37-41, 77-79, 111-114, 114-118. Cite all the references in the hierarchical order.

Author Response

Point-by-point response letter

Reviewer 2:

Most of the unnecessary cited references are not required in the manuscript. Authors need to remove references cited unnecessary like 37-41, 77-79, 111-114, 114-118. Cite all the references in the hierarchical order.

Response:

We agree with the reviewer that some, but not all of the mentioned references, are partially redundant and not absolutely required.

We have removed the citations 37-41 from line 234. Ref. 41 however is a key publication in another context as well, and is still called in the manuscript, now as ref. 74 in chapter 4.3.

References 77 and 78 (now 72 and 73) are seminal studies, demonstrating that melanoma-derived EVs are enriched in the wound exudate / seroma collected from melanoma patients undergoing lymphadenectomy. These findings suggest that like in mice, tumor-derived EVs are enriched in tumor-draining lymph in cancer patients. We cannot agree to remove these references.

We have removed partially redundant references 112-116 and 118, but maintained reference 111 (now 107) and 117 (now 108). We feel that these minimal references are necessary to support the statements in our manuscript text.